# Deep Learning Approaches in Histopathology

**DOI:** 10.3390/cancers14215264

**Published:** 2022-10-26

**Authors:** Alhassan Ali Ahmed, Mohamed Abouzid, Elżbieta Kaczmarek

**Affiliations:** 1Department of Bioinformatics and Computational Biology, Poznan University of Medical Sciences, 60-812 Poznan, Poland; 2Doctoral School, Poznan University of Medical Sciences, 60-812 Poznan, Poland; 3Department of Physical Pharmacy and Pharmacokinetics, Faculty of Pharmacy, Poznan University of Medical Sciences, Rokietnicka 3 St., 60-806 Poznan, Poland

**Keywords:** artificial intelligence, image analysis, deep learning, machine learning, pathology, tumor morphology

## Abstract

**Simple Summary:**

Artificial intelligence techniques have changed the traditional way of diagnosis. The physicians’ consultation decisions can now be supported with a particular algorithm that is beneficial for the patient in terms of accuracy and time saved. Many deep learning and machine learning algorithms are being validated and tested regularly; still, only a few can be implemented clinically. This review aims to shed light on the current and potential applications of deep learning and machine learning in tumor pathology.

**Abstract:**

The revolution of artificial intelligence and its impacts on our daily life has led to tremendous interest in the field and its related subtypes: machine learning and deep learning. Scientists and developers have designed machine learning- and deep learning-based algorithms to perform various tasks related to tumor pathologies, such as tumor detection, classification, grading with variant stages, diagnostic forecasting, recognition of pathological attributes, pathogenesis, and genomic mutations. Pathologists are interested in artificial intelligence to improve the diagnosis precision impartiality and to minimize the workload combined with the time consumed, which affects the accuracy of the decision taken. Regrettably, there are already certain obstacles to overcome connected to artificial intelligence deployments, such as the applicability and validation of algorithms and computational technologies, in addition to the ability to train pathologists and doctors to use these machines and their willingness to accept the results. This review paper provides a survey of how machine learning and deep learning methods could be implemented into health care providers’ routine tasks and the obstacles and opportunities for artificial intelligence application in tumor morphology.

## 1. Introduction

Artificial intelligence (AI) was a term introduced in the 1950s by McCarthy et al. [1], describing a field in computer science that emulates human intelligence by computers designed to think and act like humans in similar situations. The concept may also allude to any device with human-like abilities, such as understanding and solving potential problems. Currently, AI provides essential tools trusted by users and makes its way into many areas of our daily lives, especially healthcare [2]. AI has a vital role in the medical field, including diagnosing skin diseases, radiology, ultrasound, and histopathology depending on image analysis technologies [3,4]. Enormous responsibilities and challenges for AI require developers to comprehend and design flexible code to overcome the complex AI algorithm, thus making it applicable to pathological diagnosis.

Prewitt and Mendelsohn [5], who pioneered visual pathology in the 1960s, took simple microscopic images of a blood smear and scanned them. These images were processed and transformed from optical data to a matrix of optical density values for image analysis. Whole-slide scanners were introduced in the late 1990s. Since then, AI-based models used in digital pathology have expanded quickly to interpret whole-slide images (WSIs) using numerous analytical methods. The construction of a wide range of digital-slide databases, such as The Cancer Genome Atlas (TCGA), allowed scientists to quickly obtain an abundant amount of selected and annotated data of pathological images connected to medical diagnosis and genetic data, paving the way for significant AI research in oncology and digital pathology [6,7]. In 2012, a team of researchers used TCGA data to discover a unified genetic and morphological pattern consistent with the response of chemotherapy treatment in ovarian cancer [8], including an elementary machine learning (ML) prototype for WSIs from TCGA.

Deep learning (DL) and ML are subtypes of AI, and the experts have defined and distinguished between the three terms for better understanding. AI refers to intelligent machines that think and act like human beings. ML refers to the systems that learn things based on previous experience and provide defined data to make the proper decision. In contrast, DL relates to machines that can think like human brains using artificial neural networks.

DL is easier to use than ML and has better accuracy, as it is suitable for a large set of data, and the input of defined features is not required as the performance improves with more data and practice (Figure 1) [9]. The continuous development of computational systems and validated algorithms has increased the number of AI-based applications. Therefore, pathologists use it broadly to prevail over the subjective visual assessment obstacles and merge other computations for more exactitude in treating tumors [10]. DL models have numerous advantages in the histopathology field, including the ability to work with unstructured data and to generate new features with high quality from datasets without human intervention, which improves the accuracy of diagnosis and leads to the optimization of the treatment protocol [11]. The multiple layers in the neural network enhance the self-learning ability while operating intensive computational tasks. Additionally, DL models utilize distributed and parallel algorithms, which effectively reduce model training time by an average of 26% and enhance the process of the cluster while maintaining the high accuracy result [12]. Barbieri et al. found in their designed algorithm for colorectal cancer detection that the developed model reduces the time of diagnosis by almost half. Moreover, the algorithm reduced the computational cost by four times less than the typical normal diagnosis process while maintaining a 94.8% higher output result [13]. Finally, DL models offer more advanced processor technology, allowing more accurate diagnostic abilities in a shorter time [14].

On the other hand, DL models have some drawbacks. They learn by practice and gradually, requiring a large dataset volume to train the model effectively. In addition, advanced learning processors require higher computational power demanding hardware with high operating abilities. In some reported cases, DL models showed highly accurate results for the training models while, at the same time, less accuracy for the real-life data. Syrykh et al. reported a 10% accuracy difference between the internal training datasets and other external practical cases in their lymphoma diagnosis model due to the lower resolution and quality of datasets and the lower accuracy of the designed model between 63–69% [15].

This paper presents a survey of recent, up-to-date AI and DL studies and an analysis of different tumor histopathology applications to determine the advantages and limitations. Moreover, we discuss the future opportunities and challenges that might arise from the cooperation between humans and machines in tumor histopathology.

## 2. Deep Learning Applications in Tumor Pathology

AI applications in tumor pathology cover nearly all types of tumors and are engaged in prognosis, diagnosis, classification, grading, and staging. AI algorithms have been designed to assess pathological attributes, genetic modifications, and biomarkers. Examples of AI applications in tumor pathology are displayed in Table 1.

### 2.1. Diagnosis of Tumor

Pathologists must differentiate cancer from healthy cells and malignant from benign tumors, and these distinctions may significantly impact clinical decisions for various therapeutic approaches. Researchers have been able to develop AI algorithms for that purpose; for instance, convolutional neural network (CNN)-based AI algorithms have been designed by Bardou et al. [64] (Figure 2).

To distinguish the WSIs of breast cancer into two groups (cancer and non-cancer) with a precision level of 83.3% and categorize the result into four groups (healthy tissue, benign lesions, cancer in situ, and invasive cancer) with 77.8% precision, a stacked CNN was first trained to identify relatively lower attributes and then used as an input dataset to build a higher level of the stacked network. This program was developed by Bejnordi et al. [17]. They could differentiate breast malignancy from typical lesions with a 0.962 of the regions under the recipient operating curve (AUC or AUROC) and characterize invasive ductal cancer, ductal cancer in situ, and benign lesions with a precision of 81.3% using this CNN model. Bejnordi et al. [18] developed an algorithm based on the CNN system to integrate known stroma attributes to differentiate benign lesions from breast cancer, taking into account the impact of stroma on tumors. Skilled pathologists and DL-based AI algorithms were able to distinguish between malignant and benign tissues of colorectal tumors [36,38], as well as skin cancer from nevi (the plural of nevus) [60]. Mercan et al. [20] categorized breast tumors as proliferative, non-proliferative, atypical hyperplasia, cancer in situ, and invasive cancer based on breast biopsy WSIs with an 81% accuracy. This was made by using weakly supervised DL models that significantly decreased the burden of labeling. With an 86.5% precision, Wang et al. [44] categorized lesions of gastric tissues into normal, dysplasia, and cancer, while Tomita et al. [59] classified esophageal tissue as cancer, dysplasia, and Barrett esophagus with an 83% precision. Pathologists should conduct cytology analysis parallel with biopsy and excision specimens as part of their regular work. In the images diagnosed based on liquid and smear samples for the cervical cytology, AI could identify cells as normal or abnormal with a precision of 98.3% and 98.6%, respectively [33]. Based on the attributes of the cell [61] or WSI level features [68], AI-based algorithms have the power to distinguish high-grade urothelial carcinoma and its suspected cases from other urine cytology. According to the cytological images, AI also demonstrated promising potential in the comparative diagnosis of thyroid tumors [57].

### 2.2. Classification of Tumor

Different subtypes of cancer have different therapeutic approaches. Images from biopsy samples, frosted tissues, and formalin-fixed paraffin-embedded (FFPE) tissues showed a high AUC (0.83–0.97) in a study that used a CNN-based algorithm to directly separate non-small cell lung cancer (NSCLC) into squamous cell carcinoma, large cell carcinoma, adenocarcinoma, and normal lung tissue [49]. Bearing in mind the divergent patterns of lung adenocarcinoma cell growth that have been linked to patient clinical results, the CNN model designed by Gertych et al. [50] and Wei et al. [51] was used to classify every single image tile considering the pattern of growth for each individual and produce a likelihood map for the WSI, making it easier for pathologists to describe the principal and malignant elements of lung adenocarcinoma, including papillary, micropapillary, solid, and acinar components, quantitatively. Cervical squamous cell carcinoma, colorectal polyp [13], thyroid tumor [58], ovarian cancer [62], and breast tumor [21] were all multi-classified using a DL-based AI. This ability allowed the AI-based models to identify the different lung cancer histological subtypes with a precision of 60% to 89% based on cytological images [48].

### 2.3. Grading of Tumor

Pathologists evaluate tumor grades mainly depending on the tumor cell variation, cell division, necrosis, glandular structure, and other contextual factors affecting treatment decisions and clinical surveillance. To determine the grade of gliomas, Ertosun and Rubin [69] designed two different CNNs: one was able to correctly classify the patients with low-grade glioma or with glioblastoma multiforme with a 96% accuracy, while the other was able to distinguish the grade II glioma from grade III with a 71% accuracy. A CNN-based algorithm correctly identified medium-, moderate-, and high-grade breast cancers in 69% of breast biopsy images [22]. With a 91% precision, pathologists have used DL-based methods effectively to distinguish between the grades of colorectal adenocarcinoma into normal tissue, low-grade, and high-grade [38]. In the prostate cancer area, AI and ML algorithms have shown accurate and promising models in the grading process of prostate cancer. Several studies found that these models can achieve pathologist-level performance. One of the famous prostate cancer competitions is the PANDA challenge, which stands for Prostate cANcer graDe Assessment using the Gleason grading system [70]. The PANDA challenge involved 12,625 whole-slide images (WSIs) of prostate biopsies from 6 different areas and engaged 1010 groups from more than 60 countries, making it the most significant histopathology competition. The challenge system proved efficient, resulting in the first team achieving pathologist-level grading performance in only ten days. The PANDA challenge, hosted on the Kaggle platform in April–July 2020, rigorously validated the top-performing algorithms across international patient cohorts. Perincheri et al. developed a model from 118 cases to detect high-grade prostatic intraepithelial neoplasia with a 97.7% sensitivity and 99.3% specificity [71]. By using 549 slides for training and 2501 slides for testing, Pantanowitz et al. developed a model with 99.7% accuracy to detect atypical small acinar proliferation (ASAP) and perineural invasion (PNI) [72]. Moreover, Ström et al. created a model for prostate cancer detection and Gleason score using 6953 biopsies for training and 1718 biopsy for testing, resulting in a model with an AUC of 0.997 [73].

### 2.4. Staging of Tumor

Pathologists should have as many details as possible about excision samples for tumor node metastases (TNM) staging to achieve the proper treatment decisions. The developed CNN-based algorithm was able to identify three categories of the region of interest (ROI) in osteosarcomas, such as a tumor, non-tumor, and necrotic portion (e.g., cartilage, bones), on the patch level (around 64,000 patches from 82 WSIs) with a precision of 92.4% [74]. Additionally, it is possible to measure the rate of necrosis, a variable element in prognosis. For that purpose, numerous DL-based models were established to identify breast cancer tumor areas [20,23,24]. Pathologists must evaluate lymph node metastasis as part of tumor staging, but unfortunately, this process consumes time, and there is a possibility of false outcomes. Two AI models outperformed the pathologists’ findings in the Cancer Metastases in Lymph Nodes Challenge (CAMELYON16). The challenge aimed to compare the performance of AI systems and human pathologists in evaluating novel algorithms that detect the metastasis of cancer cells to lymph nodes in breast cancer. In slide-level diagnosis (recognizing whether cancer metastasis has existed), the best model achieved an AUC of 0.994.

Moreover, another two algorithms surpassed pathologists’ skill in detecting the level of lesions (identifying all metastases without discrete tumor cells) with the best mean accuracy obtained over six false-positive rates of 0.807 [25]. Furthermore, using the same dataset and sorting out artifacts, the more efficient algorithm, Lymph Node Assistant (LYNA), obtained a better AUC and sensitivity with values of 0.996 and 91%, respectively. It also revised and fixed two slides the producers had incorrectly diagnosed as “natural” [26]. Finally, the detection of micro-metastases in lymph nodes was significantly improved using LYNA, with the average accuracy increased by 8% (*p* = 0.02) to obtain 91% instead of 83% for all samples with a slightly faster assessment period [27].

In the last decade, several studies revealed that circulating tumor cells (CTCs) could be potential determinants in estimating cancer cells’ growth and development in metastatic [75,76], even with cancer patients at the early stages [77]. CTC counts above a certain threshold are linked to serious illness, heightened metastasis, and a shorter time to relapse [78]. CTCs are intended for use as a tool to measure tumor growth and facilitate clinical treatment, along with signaling treatment success, due to the ease and limited intrusion of blood collection [79]. Nevertheless, hindrances in technical matters, including limited supply and shortage of standard assays for detection and validated markers, hinder its therapeutic use [80]. According to Zeune et al. [81], DL-based CTC detection was comparatively stable with better precision than usual human opinions. In contrast, human reviewers and counting programs differed in their manual counting of CTCs from NSCLC and prostate cancer using images with fluorescence. Considering AI’s current role in recognizing tumor areas, identifying lymph node metastasis, and detecting CTCs, as well as its ability to process vast quantities of data, AI models could assist pathologists and oncologists in the process of tumor staging.

### 2.5. Assessment of Pathological Attributes

A tumor cell’s tendency to multiply is represented by mitosis. Though, counting mitosis takes time. Therefore, an effective algorithm was generated in the Assessment of Mitosis Detection Algorithms 2013 (AMIDA13) challenge to identify the mitoses of breast cancers at high-power fields (HPFs) with an 0.611 F1 score using 1000 images that could be compared to the inter-observer agreement [28] known protein structure. The Tumor Proliferation Assessment Challenge 2016 (TUPAC 16) [30] published breast cancer proliferation scores based on WSI-level AI recognition. Tumor budding is considered an offensive behavior of tumors; therefore, its analysis is crucial. Weis et al. [40] used CNN-based models to calculate the actual figure of tumor budding in cases of colorectal carcinoma. Moreover, they could determine the association between the hotspot and lymph node conditions. The type and quantity of tumor-penetrating immune cells have been linked to immunotherapy susceptibility and diagnostic stratification in cancer patients [82,83]. In breast cancer, a DL approach with a cluster of differentiation (CD)45 marked digital images could measure immunity cells and differentiate between areas rich in immune cells and regions poor in immune cells [31]. Therefore, one of the DL-based AI advantages is the ability to identify and recognize domain-agnostic and hand-crafted attributes that could be used in different diseases and types of tissues [52].

### 2.6. Assessment of Biomarkers

The DL-based model was designed by Saha et al. [29] to identify high proliferation areas and measure the severity of cancer metastasis in breast cells using the Ki-67 scale. In contrast, an AI-based model was designed by Vandenberghe et al. [32] to segment both interstitium and normal pancreatic tissues from tumor regions on uneven Ki-67 immunoreactive WSIs to calculate the severity of pancreatic tumors accurately, especially in neuroendocrine cells using the Ki-67 index. Moreover, several biomarkers match the patient profile with the adequate therapeutic regimen. Trastuzumab is a monoclonal antibody (Herceptin) used in treating gastric and breast cancer according to the human epidermal growth factor receptor 2 (HER2) condition. A CNN-based model with pathologist assistance achieved an average accuracy of 83% in determining the status of HER2 [84]. However, the results improved after dividing the cell membranes as the natural expression position of HER2. Likewise, in gastric cancer, an AI-based model was designed to evaluate HER2-negative regions (0 and 1+), HER2-positive regions (2+ and 3+), and regions with no tumor at all with 69.9% precision [45]. An AI-based model could detect the presence of programmed death-ligand 1 (PD-L1; positive or negative) by using hematoxylin and eosin (H&E)-stained images of adenocarcinoma or squamous carcinoma lung cancers with an AUC of 0.80. The result was reasonable compared to pathologist assessments depending on PD-L1 immunohistochemistry images to identify possible patients who may have sensitivity to pembrolizumab medication [13]. A DL-based AI model evaluated biomarkers engaged in the prognosis, diagnosis, and prediction of drug interactions depending on immunohistochemical dye or fluorescent dye WSIs and HE dye WSIs.

### 2.7. Assessment of Genetic Modifications

During WSI analysis, morphological variations are examples of fundamental genetic changes. Schaumberg et al. [56] used a group of 177 patients diagnosed with prostate cancer from the TCGA, 20 of them had mutant speckle-type POZ protein (SPOP), to train several groups of the CNN model to determine whether a mutation occurred in the SPOP gene of prostate cancer or not. Then the obtained results could be validated and confirmed based on an independent cohort from MSK-IMPACT. Furthermore, since the SPOP gene mutation and TMPRSS2-ERG gene fusion are strictly incompatible [85], the estimation of SPOP mutation status offered indirect knowledge about TMPRSS2-ERG. Thus elucidating the importance of determining the SPOP gene mutation condition and its potential contribution to targeted therapy accuracy. Using the lung adenocarcinoma pathological images from TCGA, Coudray et al. [49] developed a DL-based model to anticipate the most common ten genes that had mutated. They pointed six of these genes (AUCs = 0.733–0.856), including epidermal growth factor receptor [EGFR], serine/threonine kinase 11 [STK11], SET binding protein 1 [SETBP1], FAT atypical cadherin 1 [FAT1], Kirsten rat sarcoma two viral oncogene homolog [KRAS], and TP53. Moreover, an AI-based algorithm was designed using the images of gastrointestinal cancer stained with H&E stains to determine microsatellite instability (MSI) or microsatellite stability (MSS) without conducting assays on microsatellite instability. The model tested 185 slides from Asian patients and showed robust snap-frozen samples and endometrial cancer with elevated AUC (0.77–0.84) [41]. They found that models tested and used on FFPE performed better than those tested on frozen and FFPE samples. A similar result appeared with colorectal cancer samples. Despite the designers mentioning that Asian patients have different histological gastric cancer than non-Asian patients, this model potentially provides beneficial immunotherapy solutions to a wide range of gastrointestinal cancer patients. It could be implemented lowly and not require testing for the tissues in laboratories to efficiently determine MSI tumors [41]. Therefore, patients with particular genetic alterations were classified using these AI-based models depending on inherent genetic-histologic associations, which assisted the medical team in providing the precise therapy regime.

### 2.8. Prognosis Prediction

Bychkov et al. [86] developed a DL-dependent approach for grouping patients into high- and low-risk classes based on images of colorectal cancer tissues stained with H&E stains. The technique achieved better results when using small tissue areas as input (hazard ratio [HR] 2.3; 95% CI: 1.79–3.03; AUC 0.69) compared with human experts (HR 1.67; 95% CI: 1.28–2.19; AUC 0.58) and WSIs (HR 1.65; 95% CI: 1.30–2.15; AUC 0.57), and it was proven to be an individual prognosis element using the multivariate Cox comparative analysis to examine hazard. In multicenter samples, Kather et al. [87] found that combined interstitium features (with lymphocytes, debris, adipose, desmoplastic stroma, and muscles) that were extracted using CNN might independently predict the survival rate and survival without relapse of colorectal cancer patients (HR = 2.29 vs. HR = 1.92, respectively), despite the stage of the clinical level. In lung adenocarcinoma [54] and glioma [47], it has been shown that DL-based models could estimate the risk of prognosis by learning and understanding histological characteristics. Kather et al. [41] designed an MSI-based model to predict overall survival in gastrointestinal cancer, the model was tried, and the results were impressive. According to the mentioned findings, AI-based models are suitable to be used as a predictor of health outcomes of cancer patients in addition to pathological diagnosis.

### 2.9. Different Algorithm Models for Tumors Detection

Many ML and DL algorithms in tumor detection are based on different ML methods such as Decision Trees (DTs), Artificial Neural Networks (ANNs), K-nearest neighbor (KNN), and Support Vector Machines (SVMs) [88]. One of these models is known as Deep Transfer Learning (TL), and a study used a bunch of grained classification approaches to detect the different types of brain tumors, including glioma and meningioma, with a model accuracy of 98.9% [89]. Another designed a CNN-based model called the Bayesian-YOLOv4 and was created to detect breast tumors with a scoring accuracy exceeding 92% in many training data [90]. Furthermore, a DL model was designed to detect liver tumors using an enhanced DL method called U-Net. This model combines DL algorithms and CT images resulting in a new algorithm known as Grey Wolf-Class Topper Optimization GW-CTO with a learning ability of 85% and an accuracy exceeding 90% [91]. Designing a multi-tasking AI algorithm that functions on multiple tumors is challenging. Therefore, to obtain satisfactory results, pathologists have to use a variety of AI-based algorithms for the entire pathological study, in which the neoplasm should be diagnosed, classified, and staged by various models of the algorithm, and a separate algorithm should evaluate the characteristic high-risk tumors. A DL-based model was designed by Couture et al. [92] to conduct several studies on images of breast cancer tissues stained with H&E. The performed tasks include identifying the histological subtype (lobular or ductal) with a precision of 94%, grading based on histological characters (low-, moderate-, and high-grade), which obtained an 82% precision, and evaluating the receptor’s condition of estrogen hormone (negative or positive) with an accuracy of 77%, in addition to classifying the relapse risk (low, moderate, and high risk) with an accuracy of 76%.

## 3. Expectations and Challenges

As shown in the previous findings, DL- and AI-based models promise to improve the quality of pathological diagnosis and the accuracy of prognosis. Nevertheless, some problems and hurdles remain in applying AI- and DL-based algorithms in tumor pathology.

### 3.1. Model Validation

Most recent AI-based models are based on small-scale datasets and images from a single center. Scientists continue to evolve methods to improve the dataset, such as spontaneous flipping and shifting, wobbling of color, and Gaussian blur [48,50,62]; however, the outcomes from single-center images are still counted as deviations. Slide preparation, scanner models, and digitalization vary from one center to another. When a CNN-based model for the detection of pneumonia was trained using data provided by one institution and then tested separately using data from another two institutions, Zech et al. [93] found a significant difference in the performance (*p* < 0.001). Therefore, the validation and testing of AI-based models must be conducted with different directions of many institutions before being used in medical practice to train the model properly with various datasets. As a result of sharing WSI reference datasets, we still can find some clear and aligned data around cancer types with labeled cancerous areas that help uniform the AI-based models’ assessment. Furthermore, specific digital slides with large-scale databases, such as TCGA, may be used as testing or validation datasets.

The generalization and reliability of AI-based models can be improved by developing systematic quality management and calibration tools, data sharing, and validation with data from different institutes. Besides that, AI-based models must be reviewed and refined regularly by specialists in pathology.

### 3.2. Algorithm Elucidation

There is always a debate about DL models and their algorithms elucidation, which is considered a barrier to the medical acceptance of AI methods [94,95]. Since DL-based models made their projection, numerous post hoc trial approaches or guided ML algorithms have to demonstrate the efficacy of results [26,27]. Nevertheless, post hoc studies of DL approaches have been questioned since they should not be needed to clarify how a DL-based algorithm operates [96]. Lately, several studies have combined DL-based algorithms with hand-crafted ML-based models to improve the biological model’s level of understanding and elucidation. DL-based models have been employed by Wang et al. [97] to classify the digital images of nuclei stained with H&E in the early stage of NSCLC before introducing a hand-crafted tool, including the inspection of nuclear structure and form to anticipate the relapse of tumor. Many techniques are required to elucidate AI models and algorithms and obtain users’ trust, especially clinicians.

### 3.3. Histopathology and Computing Model

The file size of histopathology slides and images are approximately 100 and 1000 times higher than that of CT images and X-rays, respectively. Consequently, high-end computer hardware with developed processors and large storage capacity is needed. A powerful AI-based model must be designed to analyze the images as an effective and robust computing and storage infrastructure. The vast bandwidth required to exchange gigapixel WSIs between servers or upload them to a cloud database and handle persistent contact networks among end-users and the cloud platforms is a challenge facing users when using cloud services. These issues will be resolved shortly due to the development of information technology infrastructure, namely the global widespread of the 5G network.

### 3.4. Pathologists’ Responsibility

Aside from the weakness of interpretation in AI, many pathologists are worried about the switch in their used procedures. Implementation of AI will force pathologists to rely on accelerated parallel processing (APP) instead of using the microscope to examine the morphology of histopathological slides. In the documentation of the diagnosis report, how will pathologists explain the AI-based diagnosis proof? When pathologists use AI to submit diagnostic information, how much burden do they bear? These problems must be addressed and solved until the collaboration between machines and humans may be applied in medical practice. Another critical problem facing pathologists is determining which algorithm or model is capable of adapting and how to standardize the performance and results from these various algorithms and models.

### 3.5. Clinicians’ Responsibility

The patients’ medical diagnosis reports helped the clinicians develop appropriate therapeutic plans. Therefore, the trust of clinicians in using AI models should be increased, accompanied by a better understanding of how this software works. The clinicians must determine the minimum required diagnostic and prognostic assays, considering the expense of the patient’s treatment. Having highly accurate results for the clinicians’ daily use is crucial. They must regularly coordinate with the AI models’ developers to address any defects or issues raised during their work.

### 3.6. Regulations

In many countries, it is required to have the patient’s consent, the physician’s accreditation, and a clarification of how the AI model works to obtain governmental approval to use the designed software in digital pathology. [98,99]. The inability to interpret AI-based methods limits their acceptance [96]. In the United States, the Food and Drug Administration (FDA) has recently begun to approve DL-based methods for therapeutic use. In 2017 [100], the Philips IntelliSite Pathology Solution obtained FDA approval, and in 2019 [101], the FDA awarded the Revolutionary Device name to the digital pathology solution PAIGE.AI [102]. The FDA has established three classes to obtain medical device certification. Class I poses the lowest risk, while the devices of Class III are the highest risk (AI-based systems have been classified as Class II or III). Although there is not yet an AI-based resolution with prediction purpose that has obtained the conformity of the European Union, Philips, Sectra, and OptraSCAN’s digital pathology solutions have earned clearance to bear such a design. Whereas the FDA seems to want to control CLIA-based processes more strictly, following the paradigm developed by CLIA-based genetic studies as a safer way for AI-based diagnostic assays to gain clinical approval.

## 4. Conclusions

Pathologists need to consider many other measurements for future diagnosis, including genomics, proteomics, and measures from multiplexed marker-staining platforms to have a detailed and clear patient profile for precise tumor therapy. Regardless of the hurdles and challenges listed above, the applications of DL-based AI for automated pathology have a promising future. The potential features of ML and DL models in digital pathology encourage clinicians to consider AI applications in medical diagnosis, as AI represents the learning capabilities enhanced by the development of algorithms and the extensive collected data. Since AI models and algorithms have been tested using many reference data and the interpretation has improved, users will have more trust in the AI. Cooperation between AI-based algorithms and pathologists will lead to precise tumor therapeutic guidance.

## Figures and Tables

**Figure 1 cancers-14-05264-f001:**
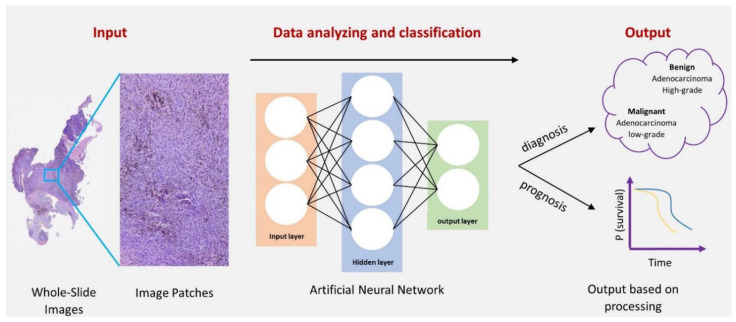
An overview of the deep learning process in pathology. Firstly, the whole-slide images (WSIs) were obtained from the original specimen slides. Then, the ongoing Artificial Neural Network (ANN) analysis process. Finally, the output of diagnosis or prognosis was based on the classification and selected features.

**Figure 2 cancers-14-05264-f002:**
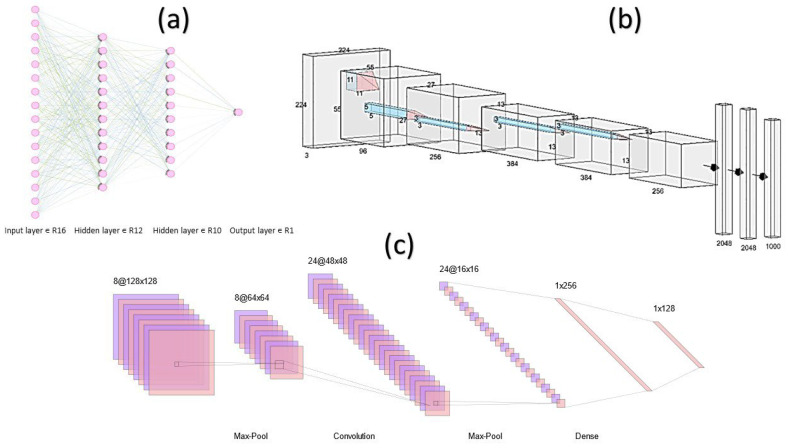
Different types of Neural Networks Architecture [65]: (**a**) Fully-Connected Neural Network (FCNN); (**b**) AlexNet is a Deep Neural Network [66]; and (**c**) LeNet refers to LeNet-5 and it is a simple CNN [67].

**Table 1 cancers-14-05264-t001:** AI applications in tumor pathology.

Training Set	AI Determinants	Outcomes	Ref.
Breast cancer			
Diagnosis			
H&E-stained images (*n* = 249; 2040 × 1536 px)	-Model I *: Carcinoma|Non-carcinoma -Model II: Normal|Benign, CIS, IC	-Model I had higher accuracy than Model II (83.3% vs. 77.8%) -Overall sensitivity = 95.6%	[16]
WSIs of H&E-stained tissue (*n* = 221; 0.243 μm × 0.243 μm)	-Model I *: Malignant|Non-malignant -Model II: Benign|DCIS, IDC	-Model I AUROC = 0.962 -Model II accuracy = 81.3%, a developing model for routine diagnostics	[17]
H&E-stained tissue (*n* = 2387; 0.455 µm × 0.455 µm)	Benign|IC	-↑AUROC = 0.962, depending only on the stromal characteristics -Estimate the amount of tumor-associated stroma and its distance from grade 3 vs. grade 1	[18]
H&E-stained biopsies (*n* = 240; 100,000 × 64,000 px; 40×) [19]	Non-proliferative|Proliferative|Atypical hyperplasia|CIS|IC	Maximum precision = 81%	[20]
Tumor subtyping			
Microscopic images (*n* = 7909; 700 × 460 px; 40–400×)	-Benign cancer: Adenosis|Fibroadenoma|Tubular adenoma|Phyllodes tumor -Malignant cancer: Ductal carcinoma|Lobular carcinoma|Mucinous carcinoma|Papillary carcinoma	Less magnification was association with better accuracy (400× = 90.66%; 200× = 92.22%; 100× = 93.81%; 40× = 93.74%)	[21]
Tumor grading			
H&E-stained breast biopsy tissue (*n* = 106)	Low, Intermediate, High	Overall accuracy: 69% -Low vs. high = 92%-Low vs. intermediate = 77% -Intermediate vs. high = 76%	[22]
Tumor staging			
Overall set (*n* = 600; validation TCGA = 200)	Regional heatmap of IC	-Dice coefficient = 75.86% -PPV = 71.62% -NPV = 96.77%	[23]
HASHI (*n* = 500) followed by testing on TCGA studies (*n* = 195)		Dice coefficient = 76%, and its analyzing power was ∼2000 in 1 min	[24]
WSIs (*n* = 270; with nodal metastases = 110) (*n =* 110)	Absence vs. presence of breast cancer metastasis in lymph nodes	-AUROC range = 0.556 to 0.994 -The algorithm performance was better than pathologists WTC [AUROC = 0.810 (0.738–0.884)***; *p* < 0.001]	[25]
WSIs of H&E-stained lymph nodes (*n* = 399 patients) [25]		-LYNA AUROC = 99% -Sensitivity = 91% at one false-positive per patient	[26]
Digitized slides from lymph node sections (*n* = 70)	Metastatic regions in lymph node	-Sensitivity = 83% and avg. processing time per image = 116 s -With algorithm-assisted pathologists, the sensitivity improved to 91% (*p* = 0.02), and the processing time reduced to 61 s (*p =* 0.002)	[27]
Evaluation of pathological features			
Mitotic figures (*n* > 1000)	Mitotic count	-IDSIA was the highest-rank approach -F1 score = 0.611	[28]
Sample images (*n =* 450; 315 training)	Ki-67 index	-GMM’s precision value = 93% -F-score of 0.91%, and 0.88% recall value	[29]
WSIs breast cancer (*n* = 821; 500 training)	-Model I: Predict mitotic scores -Model II: Predict the gene expression based on PAM50 proliferation scores	-Model I’s κ score = 0.567 (95% CI: 0.464, 0.671) -Model II’s R-value = 0.617 (95% CI: 0.581 0.651)	[30]
A set of super px images (*n* = 123,442)	-Model I: Identify regions of immune cell-rich and immune cell-poor -Model II: Quantify immune infiltration	-Model I, CNN’s F-score of 0.94 (0.92–0.94) *** -Model II, only 200 images were used, and the CNN was compared to pathologists and achieved a similar agreement level of 90% with κ values of 0.79 and 0.78	[31]
Evaluation of biomarkers			
A cohort of breast tumor resection samples (*n* = 71)	HER2 status: Negative|Equivocal|Positive	-Overall accuracy = 83% (95% CI: 0.74–0.92) -Cohen’s κ coefficient = 0.69 (95% CI: 0.55–0.84) -Kendall’s tau-b correlation coefficient = 0.84 (95% CI: 0.75–0.93)	[32]
**Cervical cancer**			
Diagnosis			
-Herlev Dataset: Abnormal and normal cell image (*n =* 100 and 280) -HEMLBC Dataset: Abnormal and normal cells (*n* = 989 and 1381) Both dataset sizes = 256 × 256 × 3 px	Normal|Abnormal	-Accuracy = 98.3% -Specificity = 98.3%. -↑AUC = 0.99. -Higher results were reproducible on the HEMLBC dataset	[33]
Tumor subtyping			
Original image group (*n* = 3012 datasets) and augmented image group (*n* = 108432 datasets), 227 × 227 px	Keratinizing|Non-keratinizing|Basaloid squamous cell carcinoma	The original images displayed significantly higher accuracy (*p* < 0.05) than the augmented group, with values of 93.33% and 89.48%, resp.	[34]
**Colorectal cancer**			
Diagnosis			
H&E-stained images (*n* = 165; 0.62 µm; 20×) [35]	Benign|Malignant	-Accuracy ≥ 95% -↑F1-score > 0.88, and the false-positive benign cases were zero	[36]
Pixel-based DNN for gland [37] trained on digitized H&E-stained images	-Model I (diagnosis) *: Normal|Cancer -Model II (grading): Normal|Low|High	Model I (diagnosis) had higher accuracy than Model II (grading), with 97% and 91%, resp.	[38]
Tumor subtyping			
Reference standard dataset (*n* = 2074)	Hyperplastic polyp|Sessile serrated adenoma|Traditional serrated adenoma|Tubular adenoma|Tubulovillous|Villous adenoma	The methodology of the residual network architecture yielded superior results in classifying the six major determinants with a value of 93.0% (95% Cl = 89.0–95.9%)	[39]
Evaluation of pathological features			
Pan-cytokeratin-stained WSI (*n* = 20)	No. tumor budding	-Spontaneously detected the absolute number of tumor buds for each image, *R*^2^ = 0.86 -Nodal status was neither associated with tumor buds at the invasive front nor the number of hotspots	[40]
Evaluation of genetic changes			
-Dataset I: Large patient cohorts from TCGA (*n* = 315) -Dataset II: FFPE samples of stomach adenocarcinoma (*n* = 360)	MSI|MSS	The AUC of dataset I (0.84, 95% CI = 0.72–0.92) was higher than the AUC of dataset II (0.75, 95% CI = 0.63–0.83)	[41]
**Gastric cancer**			
Diagnosis			
H&E-stained images (*n =* 606; 0.2517 μm/px; 40×)	Normal|Dysplasia|Cancer	RMDL = 0.923, good accuracy of 86.5%. The outcomes of this method were better than those implemented by MISVM [42] and Attention-MIP [43] with values of 0.908, 82.5%, and 0.875, 82%, resp.	[44]
Evaluation of genetic changes			
Original uncropped images (*n* = 21,000) were used to produce testing dataset (*n =* 231,000) and for detection of necrosis (*n =* 47,130)	HER2 status: Negative|Equivocal|Positive	The CNN approach had higher performance detecting necrosis than the overall HER2 classification with values of 81.44% and 69.90% resp.	[45]
**Glioma**			
Tumor grading			
Digitized WSIs obtained from TCGA	-Lower-grade glioma: Grade II|Grade III -Glioblastoma multiforme: Grade IV	-CNN distinguished lower-grade glioma from glioblastoma multiforme with accuracy = 96% -Grade II and Grade III classification accuracy lowered to 71%	[46]
Prognosis prediction			
Dataset obtained from TCGA (*n =* 769)	Risk: Low|Intermediate|High	The prognostic power of SCNN median c index = 0.754, and it was comparable with manual models, median c index = 0.745, *p* = 0.307	[47]
**Lung cancer**			
Tumor subtyping			
Multiple images (*n* = 298; 2040 × 1536 px; 40×)	-Model I **: Small and non-small cell cancer -Model II: Adenocarcinoma|Squamous cell|Small cell carcinoma	-Model I had a substantial accuracy of 86.6%, and it was higher than Model II with an overall accuracy of 71.1% -The lowest accuracy rate was in the determination of squamous cell carcinoma, with a value of 60%, while the highest was for adenocarcinoma, with a value of 89% -The accuracy of small cell carcinoma was moderate at a value of 70.3%	[48]
WSI dataset obtained from Genomic Data Commons database (*n =* 1635)	-Model I: Adenocarcinoma|Squamous cell carcinoma -Model II (gene prediction): STK11|TP53|EGFR|SETBP1|KRAS|FAT1	-Model I performance was high (AUC = 0.97) to classify the three subtypes -Six out of ten of the most mutated genes were predicted, AUC = 0.733–0.856 ***	[49]
Image tiles (*n =* 19,924) obtained from 78 slides from two institutions: CSMC and MIMW	Solid|Micropapillary|Acinar|Cribriform|Non-tumor	Overall, slides from CSMC had higher quality; their accuracy level was significantly higher (*p* < 2.3 × 10^−4^) than MIMW with values of 88.5% and 84.2%, resp. Overall accuracy in differentiating the five classes was 89.24%	[50]
Digitized WSIs (*n =* 143)	Lepidic|Solid|Micropapillary|Acinar|Cribriform	-The results were compared with a group of pathologists (*n* = 3), with κ score of 0.525 and an agreement of 66.6% -The performance was marginally higher than the inter-pathologist κ score of 0.485 and agreement of 62.7%	[51]
Dataset obtained from NCTD Tissue Bank (*n =* 39) stained for markers CD3, CD8, and CD20 and stained all T-cells, cytotoxic T cells, and B-cells, resp.	Immune cell count	-The accuracy of the augmented patch level was 98.6% -The stained tissues with T-cells were successfully classified with a sensitivity of 98.8% and specificity of 98.7% -The false-positive and false-negative detection rates were 1.30% and 1.19%, resp.	[52]
Evaluation of biomarkers			
Training set (*n* = 130 patients; training = 48)	PD-L1 status: Negative|Positive	-AUROC = 0.80, *p* < 0.01, and it persisted effectively over a range of PD-L1 cutoff thresholds (AUROC = 0.67–0.81, *p* ≤ 0.01) -AUROC was slightly decreased when dissimilar proportions of the labels were randomly shuffled for simulating inter-pathologist disagreement (AUROC = 0.63–0.77, *p* ≤ 0.03)	[53]
Prognosis prediction			
Independent patient cohort (*n* = 389)	Risk: Low|High	-The predicted low-risk group had better survival than the high-risk group (*p* = 0.0029) -It serves as an independent prognostic factor (high-risk vs. low-risk, HR = 2.25, 95% CI: 1.34–3.77, *p* = 0.0022)	[54]
**Prostate cancer**			
Tumor grading			
A discovery cohort (*n* = 641 patients) and independent test cohort (*n* = 245 patients)	Gleason scoring	The inter-annotator agreements between the model and each pathologist, quantified via κ score of 0.75 and 0.71, resp., compared with the inter-pathologist agreement (κ = 0.71)	[55]
Evaluation of genetic changes			
H&E-stained slides from TCGA cohort (*n* = 177)	*SPOP* mutation|*SPOP* non-mutant	-AUROC = 0.74 -Fisher’s Exact Test *p* = 0.007	[56]
**Thyroid cancer**			
Diagnosis			
Original image dataset (*n =* 279)	Model I **: PTC|Benign nodules	The accuracy of VGG-16 and Inception-V3 in the test group was 97.66% and 92.75%, resp.	[57]
Tumor subtyping			
Fragmented images (*n =* 11,715; training = 9763)	Normal tissue|Adenoma|Nodular goiter|PTC|FTC|MTC|ATC	Both MTC and nodular goiter had an accuracy of 100% and decreased gradually: 98.89% for FTC, 98.57% for ATC, 97.77% for PTC, 92.44% for adenoma, and 88.33% for normal tissue	[58]
**Miscellaneous Applications**			
Diagnosis for esophageal lesion			
WSIs with high resolution (*n* = 379)	Barrett esophagus|Dysplasia|Cancer	The DL model accuracy = 0.83 (95% CI = 0.80–0.86)	[59]
Diagnosis for melanocytic lesion			
H&E-stained WSIs (*n =* 155) were used to extract pathological patches (*n =* 225,230)	Nevus|Aggressive malignant melanoma	-The result of the model differed from the extracted patches and WSIs since the latter had higher sensitivity, specificity, and accuracy (94.9%, 94.7%, and 95.3% vs. 100%, 96.5%, and 98.2%, resp.). -WSIs had a higher AUROC value [0.998 (95% CI = 0.994 to 1.000) vs. 0.989 (95% CI = 0.989 to 0.991)]	[60]
Diagnosis of urinary tract lesion			
WSIs of liquid-based urine cytology specimens (*n* = 217)	Risk: Low|High	Sensitivity of 83% with a false-positive rate of 13% and AUROC of 0.92	[61]
Subtyping for ovary cancer			
H&E-stained tissue sections of ovarian cancer obtained from FAHXMU (*n =* 85; 1360 × 1024 px)	Serous|Mucinous|Endometrioid|Clear cell carcinoma	Two models were designed based on the training of the original images (*n* = 1848) and augmented images (*n* = 20,328) The accuracy of the model increased from 72.76% to 78.20% when utilizing the augmented images as training data	[62]
Biomarker for pancreatic neuroendocrine neoplasm			
A set of WSIs (*n* = 33)	Ki-67 index	The DL model employed 30 high-power fields and had a high sensitivity of 97.8% and specificity of 88.8%	[63]

Abbreviations: ATC—anaplastic thyroid carcinoma; Attention-MIP—attention-based deep multiple instance learning; AUC—area under the curve; AUROC—area under the receiver operating characteristic curve; CIS—carcinoma in-situ; CNN—convolutional neural networks; CSMC—Cedars-Sinai Medical Center; DCIS—ductal carcinoma in-situ; DNN—deep neural network; FAHXMU—First Affiliated Hospital of Xinjiang Medical University; FFPE—formalin-fixed paraffin embedded; FTC—follicular thyroid carcinoma; GMM—gamma mixture model; H&E—hematoxylin and eosin; HASHI—high-throughput adaptive sampling for whole-slide histopathology image analysis; HEMLBC—People’s Hospital of Nanshan District; Herlev university hospital; IC—invasive carcinoma; IDC—invasive ductal carcinoma; IDSIA—Istituto Dalle Molle di studi sull’intelligenza artificiale; MIMW—Military Institute of Medicine in Warsaw; MISVM—multiple-instance support vector machines; MSI—microsatellite instability; MSS—microsatellite stability; MTC—medullary thyroid carcinoma; NCTD—National Center for Tumor Diseases; NPV—negative predictive values; PPV—positive predictive values; PTC—papillary thyroid carcinoma; px—pixels; RMDL—recalibrated multi-instance deep learning method; s—seconds; SCNN—survival convolutional neural networks; WOTC—without time constraint; and WTC—with time constraint. * binary model. ** cytology. *** data represented as (range).

## Data Availability

Not applicable.

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
