# Peer review of "Deep Learning Approaches in Histopathology"

_cancers, 2022, doi:10.3390/cancers14215264_

Round 1

Reviewer 1 Report

The manuscript "Deep learning approaches in histopathology" aims to explore the current applications of deep learning and machine learning in histopathology and to explore new AI applications in tumor pathology. The manuscript is interesting and with clinical impact, since digital pathology and AI are in development in clinical laboratories. The review is well structured and summarizes the majority of the studies  that associate diagnosis in different tumors with pathological tumour features and biomarkers.

In the section 2.9 Different algorithm models for tumors detection, the authors could mention some important algorithms that are already in use for cancer diagnose and stratification.

Author Response

We really appreciate your kind words and valuable comments. Therefore, we added some of the most recent and highly accurate algorithms used in tumor detection as per your recommendation here:

Many ML and DL algorithms in tumor detection are based on different ML methods like Decision Tree DTs, Artificial Neural Network ANNs, K-nearest neighbor (KNN), and Support Vector Machine SVMs [124]. One of these models is known as Deep Transfer Learning TL, which is a bunch of grained classification approaches used to detect the different types of brain tumors, including glioma and meningioma, with a model accuracy of 98.9% [125]. Another designed CNN-based model called the Bayesian-YOLOv4 was created to detect breast tumors with a scoring accuracy exceeding 92% in many training data [126]. Furthermore, there is a DL model designed to detect liver tumors based on an enhanced DL method called U-Net. This model is a combination of DL algorithms and CT images resulting in a new algorithm known as Grey Wolf-Class Topper Optimization GW-CTO with a learning ability of 85% and accuracy exceeding 90% [127].

Reviewer 2 Report

The manuscript by Ali Ahmed et al. entitled, “Deep learning approaches in histopathology“ is aimed at reviewing recent advances in the deep learning application of histopathology. Although this is an emerging and dynamic area of research, this review may be required for some corrections. In fugure 1,  it is better to add history of each AIs. In addition, it is better to add a figure about CNN structure. Further, it is better to summarize advantage(s)/disadvantage(s) for deep lrarning in  histopathology.

Author Response

Thank you for your comments. Please note that Figure 1 has been removed as per the editor’s recommendation. However, we added a new Figure 2 about CNN structure as requested. Moreover, a summary paragraph about DL advantages and disadvantages has been added in the introduction section, as below:

The deep learning models have numerous advantages in the histopathology field, including; (I) The Ability to generate new features with high quality from datasets without human intervention, which improves the accuracy of diagnosis and leads to optimization of the treatment protocol [11], (II) The ability to work with unstructured data, (III) The multiple layers in the neural network enhancing the self-learning ability while operating intensive computational tasks, (IV) Deep learning models utilize distributed and parallel algorithms, which effectively reduce model training time by an average of 26% and enhance the process of the cluster while maintaining the high accuracy result [12], (V) Cost-effectiveness as once the model has been trained it will save money and cut unnecessary expenses, [13] found in their designed algorithm for colorectal cancer detection, the developed model reduces the time of diagnosis by almost the half. Moreover, the algorithm reduced the computational cost by four times less than the typical normal diagnosis process while maintaining a 94.8% higher output result, (VI) Deep learning models offer more advanced processor technology, allowing more accurate diagnostic abilities in a shorter time [14]. On the other hand, deep learning models have some drawbacks, including (I) It is known that deep learning models learn by practice and gradually. Therefore the models require a large volume of the dataset in order to train the model effectively, (II) As the deep learning processors are more advanced, it also requires higher computational power demanding hardware with high operating abilities, (III) In some reported cases, deep learning models showed highly accurate results in the training models, while there was less accuracy in the real-life data [15] reported that in their designed model for lymphoma diagnosis, the model used was resulting 10% accuracy difference between the internal training datasets and other external practical cases. However, they claimed the reasons behind that were lower resolution and quality of datasets in addition to the lower accuracy of the designed model between 63% - 69%.

Reviewer 3 Report

This review demonstrates the specific fields where AI-based tools in digital pathology can be successfully applied. The article contains the information about different opportunities covered either machine learning or deep learning approaches. The authors made efforts to cover as many tumors and localizations as they could so some important and interesting details did not be described. The table 1 is poorly organized and too large so it should be redone to let the readers clearly understand the key details of AI application for each tumor. In summary, the article is pretty interesting and is able to fill the gap about AI in histopathology even for the reader without any knowledge about this topic.  

Author Response

We really appreciate your kind words and valuable comments. Please note that Table 1 has been reorganized and has become shorter by almost 50% as per your request.